# The Influence of the Apple Vinegar Marination Process on the Technological, Microbiological and Sensory Quality of Organic Smoked Pork Hams

**DOI:** 10.3390/foods12081565

**Published:** 2023-04-07

**Authors:** Anna Łepecka, Piotr Szymański, Anna Okoń, Urszula Siekierko, Dorota Zielińska, Monika Trząskowska, Katarzyna Neffe-Skocińska, Barbara Sionek, Katarzyna Kajak-Siemaszko, Marcelina Karbowiak, Danuta Kołożyn-Krajewska, Zbigniew J. Dolatowski

**Affiliations:** 1Department of Meat and Fat Technology, Prof. Waclaw Dabrowski Institute of Agriculture and Food Biotechnology—State Research Institute, 02-532 Warsaw, Poland; piotr.szymanski@ibprs.pl (P.S.); anna.okon@ibprs.pl (A.O.); urszula.siekierko@ibprs.pl (U.S.); zbigniew.dolatowski@ibprs.pl (Z.J.D.); 2Department of Food Gastronomy and Food Hygiene, Institute of Human Nutrition Sciences, Warsaw University of Life Sciences-SGGW, 02-776 Warsaw, Poland; dorota_zielinska@sggw.edu.pl (D.Z.); monika_trzaskowska@sggw.edu.pl (M.T.); katarzyna_neffe_skocinska@sggw.edu.pl (K.N.-S.); barbara_sionek@sggw.edu.pl (B.S.); katarzyna_kajak_siemaszko@sggw.edu.pl (K.K.-S.); marcelina_karbowiak@sggw.edu.pl (M.K.); danuta_kolozyn_krajewska@sggw.edu.pl (D.K.-K.)

**Keywords:** meat products, marinate, uncured, organic meats, microbial assessment

## Abstract

The effect of marinating pork hams in apple vinegar on the technological, microbiological, and sensory quality was verified. Three variants of pork hams were produced: S1—ham with curing salt, without apple vinegar; S2—ham with curing salt and 5% apple vinegar; S3—ham with salt and 5% apple vinegar. The tests were carried out immediately after production, after 7 and 14 days of storage. The products did not differ significantly in their chemical composition, salt content, fatty acid composition, and water activity (*p* > 0.05). During storage, a significant increase in the cholesterol content was observed (64.88–72.38 mg/100 g of the product). The lowest levels of nitrites and nitrates were recorded for treatment S3 (<0.10 and 4.73 mg/kg of product, respectively). The samples with the addition of apple vinegar (S2 and S3) were characterized by a lower pH value, higher oxidation-reduction potential, and TBARS (thiobarbituric acid reactive substances). Hams S3 were significantly brighter (L* 68.89) and less red (a* 12.98). All of the tested pork hams were characterized by very good microbiological quality (total number of microorganisms, number of lactic acid bacteria, number of acetic bacteria, number or presence of pathogenic bacteria). Significantly the lowest TVC (total viable counts) was found in the ham S3 (2.29 log CFU/g after 14 days). The S3 hams during storage were characterized by greater juiciness (6.94 c.u.) and overall quality (7.88 c.u.), but a lower intensity of smell and taste than the cured ham (S1). To sum up, it is possible to produce pork hams without the addition of curing salt, using natural apple vinegar as a marinade. Apple vinegar has a positive effect on the storage stability of the products, without losing their sensory properties.

## 1. Introduction

Vinegar has been used for centuries to preserve food. In addition, due to its sour taste, it is used to enrich the sensory qualities of food, as an ingredient in marinades or for its medical properties. The health-promoting properties of vinegars are related to the high content of nutrients and bioactive ingredients as well as antioxidant activity [1].

Apple vinegar, next to wine vinegars and alcoholic vinegars, is one of the three most popular vinegars available on the European Union market. Given the availability of the raw materials, apple vinegar is often traditionally produced by households for subsistence. This two-stage spontaneous process is carried out by the native microbiota present in and on the fruit, with the minimal technological intervention. A common practice in the production of vinegar is the use of starters, due to the reduction in fermentation time, ensuring repeatability and the appropriate quality of the finished product [2,3,4,5]. Apple vinegar has GRAS status (*Generally Recognized As Safe*) and is very rich in organic acids, phenolic compounds, tannins, flavonoids, and carotenoids, which gives it antioxidant and antibacterial properties against many pathogens [1,6,7,8,9].

The addition of fruit vinegar to meat products has been practiced for many years. The preparation of a seasoning mixture containing salt, vinegar, sugar, oil, and spices, and placing the meat for a few or several days, allows to improve the quality characteristics of the meat product [10,11]. A number of benefits from the use of vinegar have been noticed, such as a positive effect on the sensory characteristics of meat, colour, increase in palatability, tenderness, increase in the amount of bioactive substances secreted from muscle tissue, microbiological safety and the improvement of product durability [12,13,14]. The tenderness of meat is affected by the pH of the marinade, which causes the swelling of muscle fibers and connective tissue and increases the extraction capacity of myofibrillar proteins. In addition, soaking meat in an acidic solution affects the water absorption, juiciness and colour of meat [15,16,17].

The Prof. Waclaw Dabrowski Institute of Agriculture and Food Biotechnology—State Research Institute (IBPRS-PIB) developed innovative technology for the production of fruit vinegar using the local cultures of microorganisms from its own collection. The production technology has been implemented in several juice processing plants [18,19]. On the other hand, the production of organic meat products has become a huge challenge for the industry. This is due to the difficulty and specificity of production, lower yields compared to the conventional system, difficulty with soil nutrient management, certification and market barriers, and the impossibility of using most additives [20]. This manuscript presents the possibilities of using fruit vinegar in the production of organic meat products. The aim of the study was to determine the effect of the addition of apple vinegar, obtained from natural fermentation, on the physico-chemical, microbiological, and sensory properties of organic smoked pork hams.

## 2. Materials and Methods

### 2.1. Materials

#### 2.1.1. Apple Vinegar

The raw material for the production of vinegar was natural, cold-pressed apple juices obtained from farms. The first stage of the production was based on the anaerobic fermentation of apple juice at 25 °C. Tokay wine yeast was used. The second stage consisted of the biosynthesis of acetic acid from the resulting apple wine with the help of acetic bacteria. The acetic bacteria *Acetobacter pasteurianus* O4 (KKP 674; GenBank accession OM200034) and *Acetobacter pasteurianus* MW3 (KKP 2997; GenBank accession OM212983) came from the IBPRS-PIB collection of industrial microorganism cultures. Biosynthesis took place at 30 °C. In the production process, apple vinegars were obtained with a strength of 3.30–4.50 g acetic acid/100 cm^3^ and an alcohol content of about 1%. The number of acetic bacteria was over 6 log CFU/mL, the pH of the product was over 3, and the vitamin C content was 0.72–0.95 mg/100 mL. The resulting apple vinegar was naturally cloudy, light yellow, and had an aromatic fruity-sour flavour and odour, with a sweet wine note, with no foreign aftertaste [18,19].

#### 2.1.2. Organic Smoked Pork Hams Production

The raw meat material for the manufacture of the smoked hams was 9 pork biceps muscle without outside fat and skin (*musculus biceps femoris*) weighing 1000 to 1100 g each. The raw material originated from organic breeding from one litter. The meat was obtained from industrial cutting carried out in the meat processing plant located in Dukla in Poland. The meat processing plant operates under the organic production system (certificate PL-EKO-01-5472). The meat was excised at 48 h post-mortem from carcasses cooled down at 2 °C. The raw material used had no quality defects; the pH of the muscles was between 5.6 and 6.0. The muscles were randomly divided into three experimental batches with three muscle each. Three different treatments for the smoked hams were produced: S1—control ham with a nitrite curing mixture (99.5% NaCl, 0.5% NaNO_2_); S2—ham with a nitrite curing mixture and apple vinegar; S3—ham with salt and apple vinegar. At first, the S1 and S2 treatments were cured using a nitrite curing mixture (1.6% *w*/*w* in ratio to the meat), the S3 treatment was salted (1.6% *w*/*w* in relation to the meat), then in the S2 and S3 treatments, cold apple vinegar was added (5% *w*/*w* in relation to the meat), in the S1 treatment, cold water was added (5% *w*/*w* in relation to the meat). All of the treatments were mixed separately by hand. The level of the addition of apple vinegar was determined on the basis of preliminary research (unpublished study). Then, all the treatments were stored in separate plastic containers at 2–4 °C for 5 days. After being stored the muscles were hung on smoking sticks on a truck and dried in a smoking and steaming chamber KWP2/G (REX-POL Sp. z o.o., Chorzów, Poland) for 60 min at a temperature of 45–50 °C, then the muscles were smoked with hot smoke 55–60 °C for 40 min. Smoking chips (10 mm) from beech wood were used. After that, the muscles were steamed at a temperature 85 °C up to reaching a final internal muscle temperature of 70 °C. After processing, each of the smoked hams (finished product) was cut into three portions, individually vacuum-packed, and stored at a temperature of 4–6 °C. Products were tested after production (time 0) and after 7 and 14 days (time 1 and 2) of chilled storage. The entire experiment was repeated three times.

### 2.2. Methods

#### 2.2.1. Chemical Composition and NaCl Content

The water content was determined according to ISO 1442:1997 [21], the nitrogen content and then the determination of the amount of protein was carried out by the Kjeldahl titration method according to ISO 937:1978 [22], the free fat content was determined by the Soxhlet method according to ISO 1444:1996 [23], the carbohydrate content was determined by the titration method according to the IBPRS-PIB laboratory’s own procedure (PA/09, Issue 3 of 8 June 2021), which is based on the PN-A-82059:1985 [24,25]. NaCl content was determined by the potentiometric method according to ISO 1841-2:1996 [26]. The values are given in %.

#### 2.2.2. Fatty Acids and Cholesterol Content

Fatty acids were determined by gas chromatography with a flame ionization detector using HP/Agilent 6890 II-FID (Hewlett-Packard; Palo Alto, CA, USA) according to ISO 12966-1:2014 [27]. A summary of the results (sum of saturated, monounsaturated, polyunsaturated, and trans, n-3, and n-6 fatty acids) is presented in the paper, and the values are given in %. Cholesterol was analysed by gas chromatography with a flame ionization detector (HP/Agilent 6890 II-FID) according to the IBPRS-PIB laboratory’s own procedure (PA/04, issue 7 of 8 June 2021) [28], and the values are expressed in mg/100 g of the product.

#### 2.2.3. Nitrite and Nitrate Content

Nitrites and nitrates were determined by high-performance liquid chromatography with spectrophotometric detection (HPLC-UV) according to PN-EN 12014-3:2006 [29] and Siu & Henshall [30]. The content of nitrate (III) and nitrate (V) anions in the tested samples was converted into NaNO_2_ and NaNO_3_ salts and is expressed as mg/kg of the product.

#### 2.2.4. Water Activity and pH Value

Water activity was determined according to ISO 18787:2017 [31] using the Aqualab Pawkit DE201 apparatus (METER Group, Inc., Pullman, WA, USA). The pH value was determined according to ISO 2917:1999 [32] using a pH-meter FiveEasy F20 with a LE438 electrode (Mettler-Toledo GmbH, Greifensee, Switzerland).

#### 2.2.5. Oxidation-Reduction Potential (ORP), TBARS (Thiobarbituric Acid Reactive Substances) Index

The red-ox potential was determined using a SevenCompactTM S220 with an InLab Redox electrode (Mettler-Toledo, Greifensee, Switzerland) according to the methodology of Okoń et al. [33]. The results are given in mV. TBARS index was determined by measuring the absorbance value of the solution and 2-thiobarbituric acid according to Pikul et al. [34]. The intensity of the colour from the reaction of 2-thiobarbituric acid with malonic dialdehyde (MDA) was measured using a U-2900 spectrophotometer (Hitachi, Tokyo, Japan) at a wavelength of 532 nm. The values are expressed in mg MDA/kg of product.

#### 2.2.6. Colour Measurement

An instrumental colour measurement was performed in the CIE L*a*b* system using a Minolta CR-300 reflectance spectrophotometer (Konica Minolta, Tokyo, Japan). The parameters were as follows: device Minolta, illuminant D65, observation angle 2°, no of reading per sample 40, aperture size 8 mm (according to Tomasevic et al. [35]). In order to calibrate the equipment, a white standard with parameters L* 99.18, a* −0.07, b* −0.05 was used.

#### 2.2.7. Microbiological Analysis

The microbiological evaluation was carried out using the plate method according to the accepted standards for food microbiology. Total viable count (TVC) was determined according to ISO 4833-1:2013 [36] on nutrient agar (Oxoid, Basingstoke, UK). The *Enterobacteriaceae* family (ENT) was enumerated according to ISO 21528-2:2017 [37] on MacConkey agar (Oxoid, Basingstoke, UK). *Escherichia coli* (EC) were enumerated according to ISO 16649-1:2018 [38] on TBX agar (Oxoid, Basingstoke, UK). Lactic acid bacteria (LAB) counts were determined according to ISO 15214:1998 [39] on MRS agar (Oxoid, Basingstoke, UK). The total number ofacetic acid bacteria (AAB) was determined by surface plate method using a modified solid medium GCA (glucose calcium carbonate agar) containing ingredients required for bacterial growth (2% glucose, 0.3% K-peptone, 0.3% yeast extract, 0.7% calcium carbonate, 2% ethanol) (Sigma Aldrich, Piekary Śląskie, Poland). Nystatin, which inhibits the growth of yeasts and molds, was also added to the medium. The inoculated media were incubated for 72 h at 25 °C, systematically assessing the growth of microbiota. The number of coagulase-positive staphylococci (*Staphylococcus aureus* and other species) (SA) was determined according to ISO 6888-1:2021 [40] on Baird–Parker agar with egg yolk tellurite (Oxoid, Basingstoke, UK). The number of bacteria was expressed in log CFU/g.

The presence of *Salmonella* spp. (SAL) according to ISO 6579-1:2017 [41] was determined on XLD agar (Oxoid, Basingstoke, UK). The presence of *Listeria* spp. including *Listeria monocytogenes* (LM) according to ISO 11290-1:2017 [42] was determined on ALOA agar (Agar *Listeria* according to Ottaviani and Agosti; BTL, Łódź, Poland).

#### 2.2.8. Sensory Analysis

The sensory evaluation was carried out using the quantitative descriptive (QDA) method in accordance with ISO 13299:2016 [43]. The evaluation team (experts) had the appropriate qualifications [44] and methodological preparation (theoretical and practical) in the field of sensory methods and extensive experience in carrying out assessments using the quantitative descriptive analysis method. The assessment was carried out in a sensory laboratory with individual stands for evaluators, in the conditions of 20 ± 1 °C, relative humidity of 40–50%, with normal light, and mechanical ventilation.

Pork ham samples in the amount of 15–20 g were placed in disposable, coded, lidded packages. A team of 10 people evaluated the pork hams in terms of odour (5 features), colour and uniformity of colour (2 features), juiciness (1 feature), flavour (8 features) and overall quality (1 feature). A scale of 0–10 c.u. was used, where 0 meant no intensity of the given feature, and 10 meant high intensity of the given feature. The results of the assessments are given as averages and expressed in c.u. The study was conducted in accordance with the ethical principles of the Declaration of Helsinki [45].

#### 2.2.9. Statistical Analysis

As part of the project, three independent production batches (replicates; n = 3) were made. Measurements were repeated several times for each production batch. The obtained results are presented as a mean and standard deviation. In order to analyse the effects (treatments, time of storage) a one-way analysis of variance (ANOVA) was performed, with a significance level of *p* < 0.05. Tukey HSD post hoc test was used to compare pairs of means. The Statistica 13.1 program (TIBCO Software Inc., Palo Alto, CA, USA) was used for the calculations.

## 3. Results and Discussion

Three types of pork hams were produced in the manufacturing processes. Table 1 presents the chemical composition and sodium chloride content in the tested hams.

Pork hams did not differ significantly in the basic composition (*p* > 0.05). The moisture content of the hams was 63.55–65.60%, the protein content was 26.85–29.05%, the fat content was 4.40–4.88% and the carbohydrate content was <0.50%. The sodium chloride content ranged from 1.63 to 2.05%. The chemical composition of the products was typical for pork hams [46] and complied with the chemical requirements for smoked meats set by the Polish standard requirement PN-A-82007:1996/Az1:1998 [47].

The tested pork hams did not differ significantly in the composition of fatty acids, both between the treatments and during storage (*p* > 0.05) (Table 2). The dominant fatty acids were monounsaturated (MUFA) and their content was over 50%. The saturated fatty acids (SFA) accounted for 36.28–37.45%, while the polyunsaturated fatty acids (PUFA) were 10.53–12.78%. The pork hams were characterized by a low content of trans fatty acids (0.10–0.20%). The initial cholesterol content was 60.20–66.30 mg/100 g of product (*p* > 0.05). After 7 days of storage, a significant increase in cholesterol content (71.25–76.05 mg/100 g of product) was found in all of the tested treatments (*p* < 0.05). The highest cholesterol values after 14 days of storage were noted for treatment S3 (after 14 days it was 72.38 mg/100 g of product, *p* < 0.05). The literature has already described a significant effect of acetic acid and/or apple cider vinegar on lowering cholesterol levels in products [48], animals [49,50,51] or humans [52,53]. Unfortunately, the products we tested had a higher cholesterol content during storage. Polyunsaturated fatty acids and cholesterol can be oxidized during the preparation and storage of meat and meat products. As a result of this oxidation, numerous compounds are formed (hydroperoxides, aldehydes, ketones, cholesterol oxides, such as oxysterols), which can contribute to an increase in cholesterol content during storage [54].

Lipid oxidation is the main factor that affects meat quality. Oxidation is common in muscle and is responsible for off-taste and discoloration in various types of meat. In addition, oxidative reactions can reduce the nutritional value of meat, and some oxidation products can be harmful to health [55]. The main fatty acids present in the tested pork hams are mono- and polyunsaturated fatty acids, which favour oxidation processes. The resulting meat products are susceptible to rancidity and rapid product spoilage. Similar results of the composition of fatty acids in conventional and traditional pork hams were obtained by Halagarda et al. [56], where the dominant fatty acids were MUFA (49.28–49.39%) and PUFA (9.00–9.70%).

The highest significant NaNO_2_ values were found in the ham S1 that was cured with (4.40–17.82 mg/kg of product) (Figure 1a). After 14 days of storage in samples S1 and S2, a significant decrease in NaNO_2_ content was observed (*p* < 0.05). Trace amounts of NaNO_2_ were found in ham S3, produced without the addition of curing salt (*p* < 0.05). The lowest NaNO_3_ values were also recorded for ham S3 (2.99–4.43 mg/kg product; *p* < 0.05) (Figure 1b). In the case of treatment S1 and S3, an increase in NaNO_3_ content was observed after 7 days, followed by a decrease after 14 days (*p* > 0.05).

Nitrites and nitrates are food additives used in meat processing. They give the appropriate red-pink colour, are resistant to high processing temperatures, improve and preserve the taste and smell, inhibit fat oxidation processes, and extend shelf life by limiting unnecessary microorganisms, such as *Clostridium botulinum*, *Staphylococcus aureus*, and *Listeria monocytogenes*. Uncured raw meat contains relatively small amounts of nitrates and nitrites [57]. While storing the hams, a decrease in the concentration of residual nitrites in the products was observed, which was associated with their reduction by microbial and meat enzymes. The increase in the content of nitrates in meat products during storage could be due to the conversion of heme dyes to nitrosyl derivatives and the dismutation of nitrite to nitrate [11,58].

The water activity of the tested hams ranged from 0.93 to 0.98 (Figure 2a). No significant differences were observed between the treatments of the tested hams (*p* > 0.05). There was a slight decrease in a_w_ during 7 and 14 days of storage, but with no significant (*p* > 0.05). High water activity (>0.90) is conducive to product spoilage, due to the possibility of the development of microorganisms [59]. Progressive proteolytic processes could have led to the production of peptides, free amino acids, amines, or amides, thus reducing the amount of water available for the development of microorganisms [60].

The highest significant pH values were observed for ham S1 (5.81–6.12) (*p* < 0.05), while the lowest pH was recorded for treatment S3 (5.30–5.61) (Figure 2b). A significant decrease in pH was observed during storage for treatments S1 after 7 and 14 days (*p* < 0.05), and for S2 after 7 days (*p* < 0.05). Similar research results were obtained by Fencioglu et al. [16]. The pH value of the steaks marinated in apple cider vinegar was approximately 5.67 compared to the control (5.90). Some studies have shown that meat dipped in acidic marinades, resulting in a pH below 5.0, has a higher water content, less loss during cooking, and is softer. It was found that dipping meat in acidic marinades induces the absorption of the marinade between the muscle fibers, which accelerates the swelling of muscle fibers and proteolytic processes [60]. In the studies of Alagöz et al. [61], it was observed that apple cider vinegar had a beneficial effect on increasing the tenderness of chicken breast meat, while significantly lowering the pH and reducing losses during cooking.

The degree of lipid oxidation is directly related to the degree of lipid unsaturation. Lipolysis, which leads to the formation of free fatty acids, is regulated by a number of specific enzymes. In this context, both the endogenous enzymes of fat cells and muscle fibers as well as bacterial enzymes play an important role in lipolysis [62]. Oxidative changes taking place in the product reduce the nutritional value, limit the shelf life, lead to sensory changes, and affect the health safety of the products. Due to the compounds formed, they can have a toxic effect on human health [63].

The oxidation-reduction potential of the tested hams immediately after production was 399.30–404.73 mV (Figure 3a). Treatments did not differ significantly from each other (*p* > 0.05). After 7 days of storage, a significant decrease in ORP was observed to the values of 339.55, 250.05 and 352.08 mV, respectively (*p* < 0.05). After 14 days, a significant increase in ORP was noted (378.08–393.85 mV; *p* < 0.05), but the treatments still did not differ significantly from each other (*p* > 0.05).

The TBARS index immediately after production and during storage remained quite low [64] (Figure 3b). After production, significantly the lowest TBARS were in ham S2 (0.54 mg MDA/kg), and the highest S3 (0.83 mg MDA/kg) (*p* < 0.05). Initially, reactions of secondary oxidation products can be inhibited by the addition of curing mixture. In turn, during storage, acetic bacteria begin to develop, which can accelerate oxidation processes. After 7 and 14 days of storage, there was a significant decrease in TBARS for ham S1 and a significant increase in TBARS in hams S2 and S3 (*p* < 0.05). Pork hams with the addition of vinegar had a significantly higher TBARS index, which may be due to the activity and metabolism of the added vinegar bacteria or lactic acid bacteria. In the case of S1 ham, the number of acetic bacteria was not found, and the addition of the curing mixture may weaken the oxidative processes. However, assuming the limitations of secondary oxidation products formed at <2.00 mg MDA/kg of product, these values do not exceed the recommended standards [64,65]. In the studies of Fencioglu et al. [16], higher TBARS values were found for vinegar-marinated steaks (0.275–0.353 mg MDA/kg) than controls (0.271 mg MDA/kg).

Meat colour remains the most important attribute of quality, and although it is not a reliable indicator of its safety and quality, it does not prevent consumers from treating it as an indicator of meat health, on which they base their purchasing decisions [35]. The production of meat products without the use of curing salt is complicated due to the difficulty in obtaining the appropriate red colour of the final product [46,66].

The tested pork hams with the addition of apple vinegar (S3) were characterized by the highest brightness parameter L* (68.35–69.26) immediately after production and throughout the storage period (*p* < 0.05) (Table 3). The darkest were hams S2 (65.29–67.50). Treatments S1 and S2 had significantly the highest shade of red (a* 15.65–16.09; *p* < 0.05). In the case of treatment S3, significant reddening was observed after 14 days of storage (a* 12.98; *p* < 0.05). All of the hams were characterized by the intensity of yellow colour, which decreased during storage (*p* < 0.05). Hams S3 were the most yellow (b* 3.12–3.98; *p* < 0.05).

The basic function of curing is to give the meat a characteristic red-pink colour and to protect its durability, especially after thermal treatment. Myoglobin is the main component of meat pigments. Nitrite, from which nitric oxide is formed, is directly involved in fixing the colour. Subsequently, nitric oxide forms with myoglobin a coloured nitrisol complex, called nitrosyl myoglobin, which responds to the formation of a red colour. Under the influence of thermal treatment, nitrosyl myoglobin is transformed into nitrosyl myochromogen, which is a stable compound [67]. Because curing salt was added in the test treatments of the S1 and S2 hams, the colour of these products was redder than hams S3. In addition, acid treatment may increase the conversion of myoglobin to metmyoglobin, which has a lower red colour intensity. Some researchers have expressed some concern about the colour change when some cuts of meat are marinated, which can reduce the appeal of the product. The colour change can be attributed to an enhanced binding reaction of myoglobin and myofibrillar protein [17,68]. In the research of Serdaroglu et al. [69], turkey breasts marinated in citric acid or grapefruit acid were also characterized by a greater brightness (L* 55.40–63.10) and a lower proportion of red colour (a* 1.20–2.30) than the control treatment. According to the authors, at a lower pH and ionic strength, muscle proteins swell and light reflection changes, resulting in a lighter muscle colour [69].

Several factors influence the activity of the microbiota. The initial microbial consortium depends on intrinsic factors (nutrients, pH, red-ox potential, buffering capacity, water activity, meat structure, and antibacterial substances) and extrinsic factors (temperature, relative humidity, and oxygen availability) of the raw meat and other ingredients added to the product (sodium chloride, nitrate/nitrite, sugars, and spices). This microbiota will be modulated throughout the technological process by temperature, time, and relative humidity, which adds value to the meat product and contributes to its sensory characteristics and to its preservation and safety [60,66,70].

The tested pork hams were of very good microbiological quality (Table 4). The total microbial count was low both immediately after production (1.10–1.42 log CFU/g) and during 14 days of storage (2.29–4.41 log CFU/g). Ham S3 had the lowest TVC(2.29 log CFU/g; *p* < 0.05). No *Enterobacteriaceae*, *Escherichia coli* and coagulase-positive staphylococci bacteria were found in the tested pork hams (<1.00 log CFU/g). Immediately after production, the number of lactic acid bacteria was 1.00 log CFU/g in all of the test treatments. After 7 and 14 days of storage, a significant increase in the number of LAB was observed for the S3 ham (2.89 and 3.43 log CFU/g, respectively; *p* < 0.05). In the samples of hams with the addition of apple vinegar and nitrite curing mixture (S2) immediately after the production process and after 7 days of cold storage, the number of acetic acid bacteria was on average 3 log CFU/g of product. On the 14th day of storage, the number of AAB decreased significantly by one logarithmic order. Such relationships were not observed in the samples of hams with the addition of vinegar and salt (S3). No growth of ABB was noted in the control trials. *Salmonella* spp. and *Listeria* spp. bacteria were not found in the tested pork hams. Despite the difficult hygienic and sanitary conditions accompanying the acquisition and processing of raw meat, the produced and stored pork hams were microbiologically safe. The high microbiological quality of meat products depends not only on the quality of the initial raw material for production, but also on the cleanliness of the premises, equipment and, above all, on the hygiene of production workers. Although the products from the treatment of S3 were not cured, their microbiological quality was higher than that of the other treatments (S1 and S2). In addition, due to the prevailing conditions (pH > 5.30, a_w_ > 0.93, high processing temperature), the risk of the development of pathogens *Clostridium* spp. and other Bacillaceae with the possibility of producing toxins is negligible [66]. Certainly, the addition of apple vinegar worked here as a protection factor. Interestingly, a significantly higher number of lactic acid bacteria was observed in S3 hams. On the one hand, the high LAB count may have further contributed to the reduction in the total viable counts (TVC), either by the production of antimicrobial substances or by competition. On the other hand, LAB could further lower the pH by producing lactic acid, which could also inhibit the development of undesirable microbiota [66]. Acetic bacteria were observed only in treatment with the addition of apple vinegar (*p* < 0.05). AAB are not a natural microbiota of fermented meat products; therefore they are not observed in the control treatment (S1). Similar observations were made by Lytou et al. [71], where marinades inhibited the growth of pathogenic microorganisms. At the same time, no inhibition of the number of lactic acid bacteria was noted, which was also observed in our own research. Organic acids have the ability to penetrate the outer membrane of Gram-negative bacteria, which results in limiting their development. The antibacterial activity of organic acids such as acetic acid is based on lowering the pH value as a mechanism of action that inhibits the growth of microorganisms. The ability to lower pH depends on the chemical properties of organic acids, such as acid constant (pKa), dissociation constant number (Ka), the concentration of undissociated species or the concentration of organic acid [6].

Several factors affect the flavour of meat, including the type of meat, parameters of the production process and meat ingredients. Fats and fatty acid composition are the main factors influencing the formation of flavour in the meat. In addition to the type and content of fatty acids, meat palatability is affected by protein content, temperature, time, water activity, reaction environment, pH value, maturation, marbling, and cooking technique [72]. Figure 4 presents the sensory evaluation of the produced hams.

Immediately after production, the dominant odours of the pork hams were cured meat and cooked meat (Figure 4a). The reddest and most uniform were the S1 hams (7.28 and 6.76 c.u., respectively). The S1 and S3 hams were the juiciest (8.28 and 7.59 c.u., respectively). The intensity of the taste varied. The flavours of cooked meat, cured meat, and salt were dominant. In the assessment of the overall quality, hams S1 (7.76 c.u.) were the best, followed by the S2 and S3 hams (6.35 and 5.88 c.u.). After 7 days of storage, the odour intensity remained similar (Figure 4b). In all treatments, a more intense red colour was noted. The S3 hams were juicier (8.03 c.u.). The intensity of the cooked meat flavour decreased, with an increase in the intensity of the cured meat (5.15–5.76 c.u.) and salty (3.92–5.34 c.u.) flavour. A decrease in overall quality was noted in all of the research treatments (5.80–7.31 c.u.). After 14 days of storage, the odour of the cured meat was dominant (Figure 4c). A less red colour of the hams was observed (4.93–7.60 c.u.). A significant reduction in juiciness was observed (4.66–6.94 c.u.), with the S3 ham being the juiciest. An increase in the intensity of the odour of cooked meat, cured meat, salty and sour was observed in all of the test treatments. The overall quality of the pork hams was 5.74–7.88 c.u., with the highest score being for the S3 hams (7.88 c.u.). In conclusion, pork hams with the addition of apple vinegar were characterized by greater juiciness and overall quality, but less intensity of odour and flavour over time. Their colour was less red, which can be an inconvenience for the consumer.

## 4. Conclusions

The proposed technology for the production of meat products may become an important element not only in the production of organic products. The obtained results, on the one hand, indicate the desirability of using apple vinegar as a factor inhibiting the growth of the total number of microorganisms, and on the other hand, reveal a source of bioactive substances. The produced organic smoked pork hams have the appropriate microbiological and sensory quality and physical and chemical parameters and are durable for storage. The addition of apple cider vinegar does not affect the chemical composition of the hams. The production of meat products without the addition of curing salt is possible, but it requires maintaining appropriate technological and hygienic conditions. Further research could concern the reduction in the amount of added NaNO_2_, with the simultaneous addition of apple vinegar.

It is worth emphasizing that the quality of uncured pork hams produced with the addition of apple vinegar was similar to pork hams with the addition of curing salt. This product deteriorated and aged slower, which is a process that is difficult to obtain in organic processing.

In organic production, the use of additional substances is limited to a minimum, which is often a major technological problem; therefore, the use of apple vinegar in organic meat processing is heading in a promising direction.

## Figures and Tables

**Figure 1 foods-12-01565-f001:**
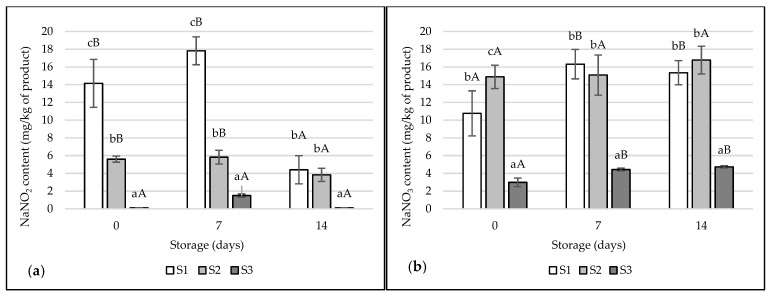
Content of NaNO_2_ (**a**) and NaNO_3_ (**b**) in the tested samples of pork hams. The values are expressed as means ± SD. Values marked with lowercase letters ^(a–c)^ differ significantly between treatments. Values marked with capital letters ^(A,B)^ differ significantly between storage times.

**Figure 2 foods-12-01565-f002:**
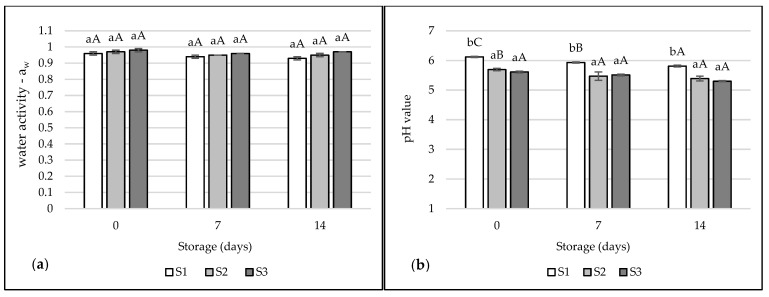
Water activity (**a**) and pH value (**b**) of the tested samples of pork hams. The values are expressed as means ± SD. Values marked with lowercase letters ^(a,b)^ differ significantly between treatments. Values marked with capital letters ^(A–C)^ differ significantly between storage times.

**Figure 3 foods-12-01565-f003:**
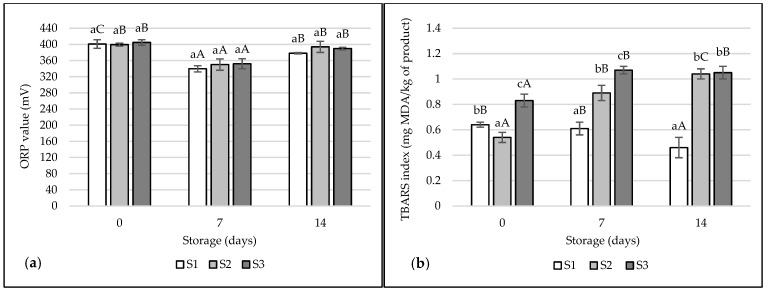
Oxidation-reduction potential (ORP) (**a**) and TBARS index (**b**) of the tested samples of pork hams. The values are expressed as means ± SD. Values marked with lowercase letters ^(a–c)^ differ significantly between treatments. Values marked with capital letters ^(A–C)^ differ significantly between storage times.

**Figure 4 foods-12-01565-f004:**
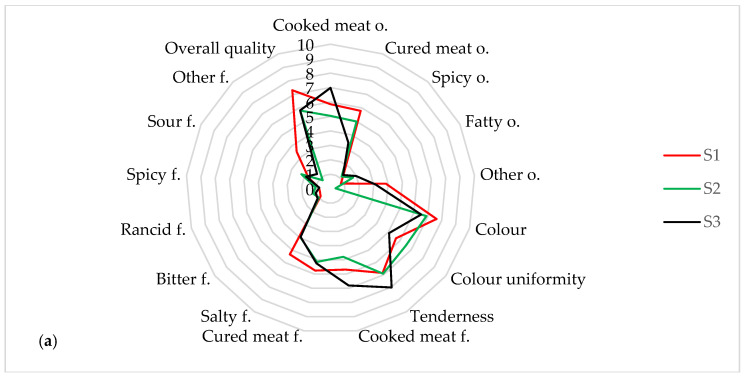
Sensory analyses of the tested pork hams immediately after production (**a**), after 7 days (**b**) and 14 days (**c**) of storage. o.—odour, f.—flavour.

**Table 1 foods-12-01565-t001:** Chemical composition and chloride content in the examined pork hams.

Treatment	Moisture (%)	Protein (%)	Fat (%)	Carbohydrates (%)	NaCl (%)
S1	65.60 ± 0.94 ^a^	26.85 ± 1.08 ^a^	4.88 ± 1.48 ^a^	<0.50	2.05 ± 0.10 ^a^
S2	63.55 ± 1.34 ^a^	28.88 ± 1.37 ^a^	4.88 ± 0.63 ^a^	<0.50	1.63 ± 0.28 ^a^
S3	64.48 ± 0.74 ^a^	29.05 ± 0.69 ^a^	4.40 ± 0.54 ^a^	<0.50	1.73 ± 0.43 ^a^

The values are expressed as means ± SD; <0.50—below the detection limit; Means in the same column followed by different lowercase letter (^a^) represent significant differences in the treatment (*p* < 0.05).

**Table 2 foods-12-01565-t002:** Fatty acid composition and cholesterol content in the examined pork hams.

Parameter	Treatment	Storage (Days)		
		0	7	14
SFA (%)	S1	36.68 ± 0.56 ^aA^	36.28 ± 0.34 ^aA^	37.45 ± 0.84 ^aA^
	S2	37.30 ± 0.70 ^aA^	37.10 ± 0.94 ^aA^	36.85 ± 0.44 ^aA^
	S3	36.50 ± 0.65 ^aA^	36.90 ± 0.94 ^aA^	37.18 ± 0.44 ^aA^
MUFA (%)	S1	50.98 ± 1.44 ^aA^	51.73 ± 1.32 ^aA^	51.98 ± 1.82 ^aA^
	S2	50.15 ± 0.66 ^aA^	50.40 ± 0.86 ^aA^	51.13 ± 1.20 ^aA^
	S3	52.35 ± 1.06 ^aA^	51.13 ± 0.86 ^aA^	49.95 ± 1.20 ^aA^
PUFA (%)	S1	12.33 ± 1.23 ^aA^	12.00 ± 1.26 ^aA^	10.53 ± 1.22 ^aA^
	S2	12.55 ± 0.66 ^aA^	12.50 ± 0.27 ^aA^	12.03 ± 1.09 ^aA^
	S3	11.13 ± 0.57 ^aA^	11.93 ± 0.27 ^aA^	12.78 ± 1.09 ^aA^
Trans (%)	S1	0.15 ± 0.06 ^aA^	0.15 ± 0.06 ^aA^	0.15 ± 0.06 ^aA^
	S2	0.20 ± 0.00 ^aB^	0.18 ± 0.05 ^aB^	0.10 ± 0.00 ^aA^
	S3	0.10 ± 0.00 ^aA^	0.10 ± 0.05 ^aA^	0.13 ± 0.00 ^aA^
Cholesterol (mg/100 g of product)	S1	62.65 ± 5.09 ^aA^	71.25 ± 1.53 ^aB^	64.88 ± 1.94 ^aA^
	S2	60.20 ± 2.53 ^aA^	73.70 ± 0.13 ^aB^	70.63 ± 3.03 ^bB^
	S3	66.30 ± 3.92 ^aA^	76.05 ± 1.13 ^aB^	72.38 ± 3.03 ^bB^

SFA—saturated fatty acids, MUFA—monounsaturated fatty acids, PUFA—polyunsaturated fatty acids, trans—trans fatty acids; The values are expressed as means ± SD. Means in the same column followed by different uppercase letters (^A,B^) are significantly different in the time (*p* < 0.05), means in the same row followed by different lowercase letters (^a,b^) are significantly different in the treatment (*p* < 0.05).

**Table 3 foods-12-01565-t003:** Colour parameters in tested pork hams.

Parameter	Treatment	Storage (days)		
		0	7	14
L*	S1	66.24 ± 2.12 ^aB^	64.63 ± 2.14 ^aA^	66.41 ± 2.55 ^bB^
	S2	65.29 ± 2.50 ^aA^	67.50 ± 2.08 ^bB^	67.17 ± 3.51 ^aB^
	S3	69.26 ± 1.46 ^bA^	68.35 ± 2.18 ^bA^	68.89 ± 2.20 ^bA^
a*	S1	16.07 ± 0.97 ^bA^	15.91 ± 1.02 ^bA^	15.65 ± 1.16 ^bA^
	S2	16.08 ± 1.46 ^bA^	15.91 ± 1.06 ^bA^	16.09 ± 1.45 ^bA^
	S3	10.81 ± 1.30 ^aA^	10.96 ± 1.27 ^aA^	12.98 ± 1.37 ^aB^
b*	S1	2.43 ± 0.52 ^aC^	2.06 ± 0.61 ^aB^	1.70 ± 0.64 ^aA^
	S2	2.13 ± 0.63 ^aB^	1.78 ± 0.68 ^aA^	2.02 ± 0.63 ^aA^
	S3	3.98 ± 1.03 ^bB^	3.75 ± 0.84 ^bB^	3.12 ± 0.99 ^bA^

The values are expressed as means ± SD. Means in the same column followed by different uppercase letters (^A–C^) are significantly different in the time (*p* < 0.05), means in the same row followed by different lowercase letters (^a,b^) are significantly different in the treatment (*p* < 0.05).

**Table 4 foods-12-01565-t004:** Microbiological quality of tested pork hams.

Parameter	Treatment	Storage (days)		
		0	7	14
TVC (log CFU/g)	S1	1.42 ± 0.39 ^aA^	1.23 ± 0.21 ^aA^	3.14 ± 0.57 ^bB^
	S2	1.10 ± 0.17 ^aA^	1.26 ± 0.24 ^aA^	4.41 ± 0.02 ^bB^
	S3	1.16 ± 0.27 ^aA^	1.32 ± 0.27 ^aA^	2.29 ± 0.06 ^bA^
ENT (log CFU/g)	S1	<1.00	<1.00	<1.00
	S2	<1.00	<1.00	<1.00
	S3	<1.00	<1.00	<1.00
EC (log CFU/g)	S1	<1.00	<1.00	<1.00
	S2	<1.00	<1.00	<1.00
	S3	<1.00	<1.00	<1.00
LAB (log CFU/g)	S1	1.00 ± 0.00 ^aA^	2.35 ± 0.05 ^aA^	2.50 ± 0.18 ^aA^
	S2	1.00 ± 0.00 ^aA^	1.10 ± 0.17 ^aA^	1.49 ± 0.19 ^aA^
	S3	1.00 ± 0.00 ^aA^	2.89 ± 0.01 ^bB^	3.43 ± 0.08 ^bB^
AAB (log CFU/g)	S1	<1.00 ^aA^	<1.00 ^aA^	<1.00 ^aA^
	S2	3.10 ± 0.10 ^aB^	3.17 ± 0.13 ^aB^	2.49 ± 0.19 ^bB^
	S3	3.24 ± 0.21 ^aB^	3.43 ± 0.08 ^aB^	3.20 ± 0.08 ^aB^
SA (log CFU/g)	S1	<1.00	<1.00	<1.00
	S2	<1.00	<1.00	<1.00
	S3	<1.00	<1.00	<1.00
SAL	S1	nd	nd	nd
	S2	nd	nd	nd
	S3	nd	nd	nd
LM	S1	nd	nd	nd
	S2	nd	nd	nd
	S3	nd	nd	nd

TVC—total viable counts; ENT—bacteria from the *Enterobacteriaceae* family, EC—*Escherichia coli*, LAB—lactic acid bacteria, SA—coagulase-positive staphylococci (*Staphylococcus aureus* and other species), SAL—*Salmonella* spp., LM—*Listeria* spp. including *L. monocytogenes*; <1.00—counts below the detection limit of the plating method; nd—not detected; The values are expressed as means ± SD. Means in the same column followed by different uppercase letters (^A,B^) are significantly different in the time (*p* < 0.05), means in the same row followed by different lowercase letters (^a,b^) are significantly different in the treatment (*p* < 0.05).

## Data Availability

The data are available from the corresponding author.

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
