# Peer review of "The Influence of the Apple Vinegar Marination Process on the Technological, Microbiological and Sensory Quality of Organic Smoked Pork Hams"

_foods, 2023, doi:10.3390/foods12081565_

Round 1

Reviewer 1 Report

The author investigated the effect of the apple vinegar marination process on the technological, microbiological, and sensory quality of organic smoked pork hams. The manuscript has some important points, and the results are readable. And the contents are suitable for this Journal. However, there are some flaws needed to revise. And the whole manuscript should be rechecked for the English writing.

1.     Line72-73, “The Prof. Waclaw Dabrowski Institute of” needs to revise to “The Prof. Waclaw Dabrowski from Institute of”?

2.     I think this introduction wants to say vinegar is important, vinegar in the meat treatment is important, and apple vinegar in this manuscript is special. If the introduction part can be more concise, the manuscript will be better.

3.     Why the TBARS index for ham S1 decreased and for hams S2 and S3 increase after 7 and 14 days of storage?

4.     If the author can present some photos of hams S1, S2 and S3, it will be more clear for readers to know the effect of apple vinegar on the color of the meat.

5.     The conclusion part also needs to be more concise.

Reviewer 2 Report

              The manuscript titled “The influence of the vinegar marination process on the technological, microbiological and sensory quality of organic smoked pork hams” highlights the advantages of apple vinegar soaking for ham production. This knowledge will be helpful for industries. The context of this manuscript does not have any problems. However, minor revisions would improve readability.

Major comment

L419–421. Pathogenic bacteria, such as Salmonella, Listeria, and Staphylococcus, were tested. They were not detected in the final products. However, their presence in the material was not described. The information can be useful to evaluate the utility of this treatment if authors can describe about it.

Minor comments

Abstract: Descriptions of the abbreviations (ORP, TBARS, and TVC) are required.

Figures: The horizontal and vertical axes must be written.

L463: “The S3 hams were juicer …”

Reviewer 3 Report

The paper is about the use of apple vinegar to marinate ham and to evaluate various parameters in comparison to the control. Although the study design is not novel, however, the descriptive study has various brought interesting aspects

1.     L24: Give full form of the term when it appears first

2.     L24: Which microbiological quality has been referred

3.     L33-35: Revise to avoid language errors

4.     L48-54: Is it related to this work?

5.     Section 2.1.1, if it’s already reported then do not provide minor details, just refer it

6.     Section 2.1.2 State if % refer to w/v or w/w ratio

7.     The observation related to increase in cholesterol due to treatments given should be candidly explained. At present, the reason given for this observation does not match with the conclusion drawn from it.

8.     The presence of enteric bacteria depends on handling person or from the raw materials used. If the raw materials do not bring these organisms, then the products would remain free from these germs. This point should be discussed here

9.     L424: Do not italicize the names other than genus and species

10.  L463: Look for a typo
